# Endo-Exoglucanase Synergism for Cellulose Nanofibril Production Assessment and Characterization

**DOI:** 10.3390/molecules28030948

**Published:** 2023-01-18

**Authors:** Ricardo Gonzalo Ramírez Brenes, Lívia da Silva Chaves, Ninoska Bojorge, Nei Pereira

**Affiliations:** 1Department of Chemical and Petroleum Engineering, Fluminense Federal University, R. Passos da Patria 156, Niterói 24210-140, RJ, Brazil; 2School of Chemistry, Federal University of Rio de Janeiro, Av. Athos da Silveira Ramos, 149, Ilha do Fundão 21941-972, RJ, Brazil

**Keywords:** nanocellulose, enzymatic synergism, optimal mixture, CCRD, Colby factor

## Abstract

A study to produce cellulose nanofibrils (CNF) from kraft cellulose pulp was conducted using a centroid simplex mixture design. The enzyme blend contains 69% endoglucanase and 31% exoglucanase. The central composite rotational design (CCRD) optimized the CNF production process by achieving a higher crystallinity index. It thus corresponded to a solid loading of 15 g/L and an enzyme loading of 0.974. Using the Segal formula, the crystallinity index (CrI) of the CNF was determined by X-ray diffraction to be 80.87%. The average diameter of the CNF prepared by enzymatic hydrolysis was 550–600 nm, while the one produced by enzymatic hydrolysis and with ultrasonic dispersion was 250–300 nm. Finally, synergistic interactions between the enzymes involved in nanocellulose production were demonstrated, with Colby factor values greater than one.

## 1. Introduction

Cellulose is transformed into nanocellulose (NC) by mechanical treatments and chemical and enzymatic hydrolysis. It can also be produced by bacterial fermentation [1]. Enzymatic hydrolysis uses cellulases to deconstruct cellulose to NC, which can be aided by automated techniques to increase its yield [2]. NC is a very appealing material due to its biodegradability, low density, transparency, low coefficient of thermal expansion, and high mechanical strength (breakdown requires pressures greater than 10,000 MPa), while materials such as steel and Kevlar need 500 MPa and 2800 MPa, respectively) [3,4]. Furthermore, NC presents a high degree of polymerization, chemical stability, biocompatibility, magnetic and electrical susceptibility, a high surface area, and protonic conductivity, among other characteristics [3,5].

The properties of NC allow its application in various areas, such as biomedicine to create drug delivery systems, biosensors, and virus elimination filters [2,6]. In the food area, it can function as a food stabilizer or as an ingredient in the formulation of food packaging [7]. It can also be used to develop high-tech energy devices such as nanogenerators, flexible transistors and others used in energy storage [8], in the formulation of hydrogels and aerogels [9,10], and as part of membrane filtration systems for sewage treatment [11].

The lignocellulosic material comprises lignin, hemicellulose, and cellulose, forming a highly stable and recalcitrant plant structure. It is necessary to submit the lignocellulosic material to different pretreatments to access the cellulosic fraction, which will later be transformed into nanocellulose through chemical hydrolysis [7,10,12]; enzymatic hydrolysis [13,14,15,16] or mechanical treatments [17].

Enzymatic hydrolysis is considered an ecologically correct mechanism for the production of NC because it does not require acid catalysts of acidic origin like that obtained by acid hydrolysis, which contributes to environmental pollution because of its corrosive nature. Enzymatic hydrolysis is considered an ecologically correct mechanism for the production of CN because it does not use acidic catalysts as when obtained by acid hydrolysis, which harms the environment due to its corrosive nature. Moreover, enzymatic hydrolysis is a technique that does not require large energy consumption, as in mechanical and thermal pretreatments [15]. Another advantage of enzymatic pretreatment is its use in deconstructing lignocellulosic material. For example, for lignin fractionation, enzymes such as laccase, manganese peroxidases, and peroxidases can be implemented; for the hydrolysis of hemicellulose, hemicellulases (mannanases, xylanases, and β-xylosidases, among others) are used and therefore, as a result of replacing acid–alkali pretreatments, cellulose structures can be better preserved, which can then be hydrolyzed to NC by using cellulases [18,19].

Figure 1 represents a schematic diagram of the production of NC derived from the enzymatic hydrolysis of kraft cellulose pulp (KCP) by two different cellulases. Endoglucanase is the most significant type of protein to produce NC. This group of enzymes breaks the cellulose polymer into shorter polymers by acting on a less organized portion of cellulose (amorphous region). Although enzymatic treatment with endoglucanase appeared to guarantee the expansion of cellulose nanofibrillation, its feasibility is limited. Few researchers have conducted experiments on the degree of synergism between exoglucanases and endoglucanases, which appears to increase cellulose hydrolysis and, consequently, nanofibrillation gradually [2]. Cellulolytic enzymes can interact and exhibit the phenomenon of synergy. Enzymatic synergies between cellulolytic entities, which take part in the cellulase complex for NC production, are generally due to the combined action of this enzymatic entity with internal and external degradation activities. The synergy between cellulases has been commonly attributed to the combined effect of enzymes with endo-lytic activity (endoglucanase) on the one hand and exoglucanase with exo-lytic activity on the other. The synergistic effect of endo–exo can occur when the endoglucanase enzyme internally cleaves the cellulose chain, creating new reducing and non-reducing ends in the cellulose fibers, which are further hydrolyzed by the exoglucanase also known as cellobiohydrolase [20].

Karim and colleagues summarized some of the ways enzyme synergism can be used to optimize cellulose hydrolysis: between two endoglucanases; within the two exoglucanases; between endoglucanase and an exoglucanase; among endoglucanase, an exoglucanase and a β-glucosidase; between a CBM and a catalytic domain; amid two catalytic domains, among both cellulosomes, and non-complex cellulases; or even between any combination of accessory proteins and some cellulase [21].

The synergy among endoglucanase, exoglucanase and β-glucosidase is considered the fundamental mechanism for the complete hydrolysis of cellulose into glucose monosaccharides [22]. Regarding exo–exo synergy, the commercial enzymes Cel7A (formerly CBH I) and Cel6A (formerly CBH II) stand out. The enzyme Cel6A hydrolyzes the non-reducing end of the cellulose molecule, whereas the enzyme Cel7A exhibits high specificity for the reducing ends of the fibers; both enzymes are from Hypocrea jecorina.

Therefore, it is preferable to use the enzyme Cel6A to hydrolyze amorphous cellulose, while the enzyme Cel7A is recommended for deconstructing crystalline cellulose domains [20].

Synergism could be “the ratio of the hydrolysis rate or yield of products released by enzymes when they act together to the sum of the rates or yield of these products when the enzymes are used separately in the same amounts as they were used in the mixture” [23]. Some research on cellulase synergism determines synergism as a ratio of sugars produced by the individual enzyme concerning the total soluble sugars produced by the individual components [24,25,26,27]. Nevertheless, other techniques exist to determine the synergism of mixtures of multiple compounds. The fundamental principle of the Colby Factor can also be implemented to assess mixtures of a different nature, such as drug mixtures, enzyme mixtures, microbial consortia, and others [28]. In the Colby Factor, the responses of active ingredients applied alone are used to calculate an expected response when these compounds are combined [29]. 

The crystallinity index (CrI) is the amount of crystalline to amorphous cellulose in a sample. CrI is a factor that contributes to biomass resilience. A high CrI indicates that the cellulosic material is highly ordered, making it difficult for cellulase to access the chains and thus affect glucose conversion [25,30,31,32]. The crystalline portion of cellulose is dense and prevents the entry of small molecules such as enzymes and water. The literature shows that cellulose crystallinity influences cellulase component synergism [33,34]. Different cellulase components have been shown to have varying cellulose adsorption capacities and activities. Banka and Mishra [35] found that crystallinity increased the adsorption of a non-hydrolytic cellulolytic component, called the fibril-forming protein of *Trichoderma reesei* cellulase enzyme-forming protein, which suggests that cellulose crystallinity greatly influences the non-hydrolytic enzymatic components required for the effective enzymatic hydrolysis of cellulose. Cellulose crystallinity may affect cellulase adsorption and the effectiveness of cellulase components that have been adsorbed.

Some researchers have reported the preparation of CNF by the enzymolysis approach together with a mechanical method: Pääkko et al. produced CNF derived from cellulose pulp (40% pine and 60% spruce) by enzymatic hydrolysis and high-pressure homogenization (HPH), obtaining CNF with a CrI of 12% [36]. Martelli et al. obtained CNF from soybean straw by enzymatic hydrolysis, HPH, and ultrasound, showing fiber lengths greater than 1 µm and a CrI of 68% [37]. Penttilä et al. produced CNF from birch pulp by fluidization and enzymatic hydrolysis, resulting in nanocellulose fibers of 5 nm in diameter and a CrI of 57% [38]. Ribeiro et al. obtained CNF from kraft cellulose pulp by enzymatic hydrolysis and ultrasound, generating fibrillar nanocellulose with diameters of 180 nm and a CrI of 78% [16].

There are many advantages to using enzymatic processes to produce CNF, whether in conjunction with other chemical treatments (sulfuric acid and TEMPO) or even solely mechanical extraction. Several authors [2,15,39,40,41] have already pointed out the following: (i) due to the high selectivity of cellulase enzymes and given that the operational conditions they require occur in milder and less dangerous conditions, allowing better control of the deconstruction process of the different solid components as well as the properties of the final material is beneficial (ii) one of the primary advantages of this control is the ability to avoid extensive/complete cellulose hydrolysis, the degree of polymerization, and the increase in the crystallinity index caused by amorphous region hydrolysis; (iii) another common advantage is a higher proportion of CNF end products; and (iv) an important aspect, if somewhat controversial, is the economic cost of enzymatic CNF production, as enzymes still have a high production cost. However, when combined with the economic benefits mentioned above and resulting from operating in mild conditions and using fewer chemicals, enzymatic treatment will facilitate both mechanical fibrillation and CNF production, reducing the energy required to produce the same amount of CNF material [42]. 

The enzymatic processes are very complex and require a synergistic study among cellulases [43] to better understand the correlation between crystallinity and enzymatic synergism. The accessibility of enzymes within the cellulose structure has been proposed as a key factor influencing enzyme hydrolysis rates [32] and, subsequently, CNF production. The accessibility of cellulases in a cellulose structure is inhibited when cellulose is mostly crystalline and is enhanced when cellulose is mainly amorphous; i.e., enzyme access is favored as a result of the greater free space available at a lower CrI [32]. Since the crystallinity of cellulose plays an important role in enzyme adsorption, it was of interest in this research to correlate enzymatic synergism with a CrI for CNF production. 

The following study was conducted to optimize an endo-exoglucanase blend to produce cellulose nanofibrils (CNF) from cellulose pulp. The enzymatic synergism of the mixture was quantified by using the Colby factor as a quantification parameter and evaluate its physicochemical and morphological properties. The crystallinity index was defined as the response variable for the future potential of using the produced CNF as a drug delivery system. In the pharmaceutical area, nanoencapsulation drugs are increasingly used to increase the number of drugs available in a specific volume. For this, the crystallinity of the material is of great interest [44].

## 2. Results and Discussion

### 2.1. Determination of Optimal Model of Enzymes Mixture

The proportion of cellulases to produce CNF was determined using the centroid simplex mixing design. A simplex centroid mixture design determined the optimal mixture of enzymes that released the lowest glucose concentration (g/L) after enzymatic hydrolysis of KCP. The experiments where nanocellulose was made at low glucose concentrations by designing a simple centroid mixture are endoglucanase and exoglucanase, which are even lower glucose concentrations than endoglucanase experiments (endoglucanase 100% pure). The centroid simplex mixture design consisted of a matrix of 10 experiments evaluating the effects of endoglucanases (EGU), exoglucanases (ExG), and β-glucosidases (BG) individually, as well as binary and tertiary mixtures of these cellulases. Statistica software (version 14.0.0.15, TIBCO) optimized the simplex centroid mixture design (for further information on the mathematical model, refer to the supplement “Determination of optimal model of enzyme combinations”). Figure 2 shows that a contour plot can assess the dependence of glucose concentration on the enzyme load employed. The green area of the graph allows one to determine the optimal point for reaching the lowest glucose concentrations, coinciding with a higher amount of endoglucanase, a minimum amount of exoglucanase, and no β-glucosidase. It was concluded that the optimal enzyme mixture corresponds to 69% EGU and 31% ExG (*p*-value < 0.05).

### 2.2. Cellulose Nanofibrils (CNF) Production

The total enzyme loading used in each central composite rotational design (CCRD) experiment consisted of EGU and an exoglucanase fraction. Both fractions are governed by the result of the optimal enzyme mixture (69% EGU and 31% ExG). From the CCRD, 13 samples were obtained (see Table 1) of which the highest crystallinity indices (CrIs) were reached for a range of enzyme loading between 0.5 U/g and 3.0 U/g and for a solid loading value between 15 g/L and 20 g/L. The optimal point that benefits the increase of the CrI of CNF was determined, corresponding to 0.974 U/g for enzyme loading and 15 g/L for solids loading (*p*-value < 0.05).

Shorter enzymatic hydrolysis times are recommended by Hu and collaborators when there is enzyme synergy involved. These same authors produced nanocellulose in a hydrolysis time of up to 3 h reaction time using endoglucanase, exoglucanases, and lytic polysaccharide monooxygenase (LPMO) within the same enzyme mixture [14].

In addition, it was decided for a constant temperature of 50 °C during the enzymatic hydrolysis of KCP to be maintained because different reports indicate this value as an optimal temperature when working with enzymes of the cellulase family: Cui and collaborators used microcrystalline cellulose from wheat fiber to produce KCP at 50 °C, derived from enzymatic hydrolysis with cellulases and aided with mechanical treatments [15]. In addition, Martelli-Tosi employed soybean straw to produce CNF by enzymatic hydrolysis with cellulases at 50 °C [37].

### 2.3. Crystal Structure by X-ray Diffraction (XRD)

The diffractograms for the KCP and CNF samples were obtained by X-ray diffraction and are shown in Appendix A. All of the KCP and CNF samples exhibited Bragg angles (2θ) expected for the cellulose I (native cellulose) diffraction peaks; the signal at the lowest 2θ was found between 14° and 17° (a complex of 1–10 and 110 diffraction); then, the signal dropped to 18° or 18.5° (amorphous area contribution) and finally, showed the maximum intensity around 22.6° (200 diffraction) [3]. The CrI calculated by the Segal equation (see item 3.5.1) is summarized in Table 1.

Some researchers establish that the Segal equation may not be accurate when finding the CrI by considering only the crystalline intensity of the largest peak, ignoring the contribution of other crystalline planes, and thus providing a CrI higher than reality [45]. The Segal method tends to give a higher CrI value than others. However, it is because the Segal equation is just an arbitrary representation of cellulose crystallinity and does not always have a physically clear meaning; a 50% CrI does not always mean half of the cellulose is crystalline and the other half is amorphous [30]. Nevertheless, it is the method most used internationally for CrI determination, which facilitates the comparison of the results obtained with other research studies that have already been executed.

KCP showed the lowest CrI values: 70.26% with the Segal method, which was expected since the biomass was not subjected to enzymatic hydrolysis; therefore, it presents a larger amorphous area compared to that of the CNF samples, which results in lower CrI values. Sample No. 7 of CNF exhibited the highest CrI: 80.87%. The CrI values determined by Segal are consistent with the CrI values already reported in the literature for NC: 78.5% by Ribeiro et al. [16], 66.4% by Viana et al. [46], and 70% according to Buzala et al. [47].

As sample No. 7 obtained the highest CrI and sample No. 11 also obtained an elevated CrI, they were the only two samples to be characterized by the following characterization techniques: scanning electron microscopy, particle size analysis, zeta potential, thermogravimetric analysis, and Fourier transform infrared analysis.

### 2.4. Scanning Electron Micrographs of Cellulose Nanofibrillar

Figure 3 displays micrographs of the KCP and CNF sample No. 7 at different magnifications generated by scanning electron microscopy (SEM). The formation of CNF with rough topography and irregular diameters along the length of the fiber can be evidenced. CNF diameters were measured using Image J software. The diameter sizes of the produced nanofibrils ranged from 50 nm to 900 nm, where the measured average was 550 nm and 600 nm. The diameters of the KCP samples varied between 5.0 µm and 7.5 µm. The CNF diameter measurements obtained previously by SEM are consistent with results published in the literature by other authors. Cui and co-workers hydrolyzed microcrystalline cellulose enzymatically with EGU, producing nanocrystalline cellulose (NCC) and, using SEM, they reported measurements within 200 nm to 500 nm [15]. Similarly, Tong and colleagues obtained NCC derived from KCP hydrolysis employing cellulases and xylanases at low enzymatic loading (10 U/mL) and, using SEM, they reported lengths between 600 nm–800 nm [13].

### 2.5. Particle Size Analysis

Table 2 shows the results of the dynamic light scattering (DLS) technique. KCP, CNF sample No. 7, and CNF sample No. 11 (which showed the highest CrI) were studied to determine the particle size in suspension. It was observed that KCP had the most increased mean hydrodynamic diameter (HD), equivalent to 509.63 nm; this is due to the fact it was not subjected to enzymatic hydrolysis, while CNF sample No. 7, with the highest CrI, showed a mean HD corresponding to 430.20 nm.

These HD data are consistent with those obtained by Ribeiro and colleagues. They created KCP-derived NCF using the commercial enzyme preparation Carezyme. They achieved HD values ranging from 405.6 nm to 562 nm [16]. Analogously, this range coincides with that reported by Cui and colleagues. They obtained HD values within 80 nm to 600 nm for several NC samples with different hydrolysis times: hydrolysis times of 72, 96, and 120 h and variable ultrasonic agitation times: 0 min, 30 min, and 60 min [15].

When comparing the HD obtained by DLS with the diameters quantified by SEM, the former provides smaller diameter sizes by undergoing ultrasonic agitation (including KCP that resulted in a diameter range between 5.0 µm–7.5 µm reached by SEM to an HD of 510 nm approximately), since it is a mechanical method that favors the defibrillation of cellulose, breaking and cleaving its bonds, thus decreasing its fibrillar size [48,49].

### 2.6. Zeta Potential

The zeta potential magnitude represents the degree of electrostatic repulsion between adjacent similarly charged particles in the dispersion. Zeta potential values of colloidal suspensions between (±10 to ±30 mV) and (±30 to ±60 mV) show initial instability and moderate stability, respectively [37]. The average zeta potential value of KCP and the CNF samples is summarized in Table 2. 

As previously described, KCP, CNF sample No. 7, and CNF sample No. 11 exhibited a stable behavior in aqueous dispersion. Their zeta potential values ranged between −40 mV and −55 mV. These zeta potential results are interesting to obtain for colloidal systems and nano-medicines, as well as particle size, which significantly impacts the properties of nano-drug delivery systems [50]. The reason for this is that the liquid surface charge, by the Derjaguin-Landau-Verwey-Overbeek theory (DLVO), also known as the stability model colloidal systems, maintains a balance between the attractive van der Waals forces and the repulsive electrical forces. If the zeta potential drops below a certain level, the emulsion droplets or colloids will clump together due to attractive forces. A high zeta potential (positive or negative), typically greater than 30 mV, on the other hand, maintains system stability. CNF sample No.11 had the highest absolute zeta potential value in this study, equivalent to −52.60 mV. KCP, on the other hand, had the lowest value (39.37 mV).

Using cellulases in NC production revealed that the buffer positively affects the nanostructure’s stability. Beltramino and co-workers reported that the zeta potential changed from −124 mV to −53 mV when changing the process of obtaining NC from acid hydrolysis to enzymatic hydrolysis with cellulases and that this change can be attributed to ion-induced modifications of the ion distribution around the CNF, provided by the buffer [51]. The zeta potential value of CNF sample No. 11 (−52.60 mV) is quite close to that achieved by Beltramino and colleagues when employing enzymatic hydrolysis (−53 mV) [51].

As mentioned before, CNF can formulate foams, emulsions, and suspensions. Depending on its surface chemistry, aspect ratio, and crystallinity, NC can control the rheology and stability of dispersions [52]; however, the zeta potential is affected by pH, temperature, and the presence of salts and impurities in the suspension. As a result, all of the above factors must be controlled to obtain reliable data. These requirements represent a significant limitation of this technique [53]. Therefore, precise protocols and techniques to characterize the dispersion, particle size, and morphology are required to ensure consistent production and application of high-quality nanocellulose products. Particle sizing techniques typically provide different but complementary information about the particle size, morphology, and degree of agglomeration. However, there is no standard method for determining the size or distribution of nanocellulose products. Moreover, it is not easy to compare data obtained from different techniques. In addition, industrial equipment is limited to a microscale nature, and most techniques for NC dimensioning are offline, time-consuming, and costly [54].

### 2.7. Thermostability Analysis

Figure 4 shows the thermal degradation of KCP, CNF sample No. 7, and CNF sample No. 11 and corresponds to the thermogram obtained by thermogravimetric analysis (TGA). The mass values were normalized to know the mass lost in the process. The three samples presented the first stage of decomposition before 100 °C, in which the absorbed water evaporation and low molecular weight components that remain on the nanocomposites’ surface occur. The loss in mass in this first stage is between 5 and 8% of the respective sample.

The second and main thermal decomposition phase occurs around 250 °C and 380 °C for NC and about 300 °C and 375 °C for KCP. A series of degradation, dehydration, and depolymerization reactions of the glycosidic units, cellulose, and hemicellulose conform to the study materials [39]. The mass lost at this point is between 55% and 75%.

NC starts to degrade around 250 °C, while KCP begins at about 300 °C; KCP degrades at higher temperatures due to the higher intensity of inter- and intramolecular interactions of cellulose and hemicellulose and due to its polymerization degree. KCP was the only one to present a slight decomposition stage above 500 °C, which follows the oxidation of carbonized residues.

### 2.8. Fourier Transform Infrared (FTIR) Analysis

Figure 5 shows the FTIR spectra of selected samples for comparison between KCP, CNF sample No. 7, and CNF sample No. 11. Overall, a strong band appeared at approximately 3400 cm^−1^, which is related to the stretching vibration of the O-H groups. The band at 2800 cm^−1^ belongs to the aliphatic CH stretching; meanwhile, the peaks at 1480 and 1250 cm^−1^ were attributed to the scissor vibration of –CH_2_; peaks at 1165–1145 cm^−1^ were associated with C-O-C asymmetric stretch vibration; and peaks between 1120–1000 cm^−1^ were attributed to stretching of the C-O bonds. Moreover, the peaks at around 910 and 890 cm^−1^ were attributed to the β-D-glucosyl group. CNF samples showed the same spectra as KCP; these results supported the conclusion that cellulose’s molecular structures remained unchanged in the cellulase hydrolysis process [4,14,15].

### 2.9. Experimental CCRD Validation

Previously, it has been stated that the optimal point of CCRD that increases the CrI of CNF corresponds to 0.974 U/g for enzyme loading and 15 g/L for solids loading. Therefore, a new set of CNF samples was produced to validate the CCRD optimal point. These samples were coded as A, B, C, D, E, F, G, and H.

The diffractograms of the validation samples are presented in Figure 6. It can be observed that the intensity of the peaks is higher for the second group of samples, i.e., for the samples that received ultrasonic treatment. The maximum intensity of the bands of the samples without ultrasonic dispersion treatment reaches a value of approximately 1700 cps. In contrast, the value corresponds to 3750 cps for the samples with ultrasonic treatment. 

Table 3 shows the calculated CrI with Segal’s equation (see item 3.5.1). Sample A had the highest CrI in the validation samples without ultrasonic treatment: 74.78%. Likewise, sample B had the highest CrI: 78.45% for the validation samples submitted to ultrasonic treatment. 

In Table 3, it is observed that the validation samples with or without ultrasonic treatment. Those who underwent ultrasonic treatment had higher CrI. For example, applying Segal’s equation for sample D reached a CrI of 72.27%. In contrast, sample C (without ultrasound) obtained a CrI of 71.93%, indicating that the ultrasonic treatment helps to increase the crystallinity and degradation of the amorphous domains of the nanocellulose. Nasir and co-workers explain that ultrasonic treatment utilizes hydrodynamic forces with oscillation power to isolate cellulose fibers by forming, expanding, and imploding microscopic gas bubbles, facilitating the access of cellulases in the cellulosic structure [55]. The ultrasonic treatment would break down the non-crystalline regions of the cellulose, thus destroying the interfibrillar bonds between the layers of the cellulose molecules and enhancing the contact area of the enzyme and substrate [15]. 

In Figure 6, the green curve—which corresponds to the enzymatic hydrolysis with pure EGU only—presents the largest crystalline peaks intensity in the diffractograms; however, this does not imply that its CrI was the largest, as one should also consider the intensity of the amorphous area of the nanocellulose, as defined in Equation (1) (see item 3.5.1). For enzymatic hydrolysis with pure EGU, the contribution of amorphous domains was also greater than the other samples, thus decreasing the value of its CrI.

Furthermore, Gibril and colleagues state that ultrasonic treatment alters the cellulose’s molecular structure due to the cavitation phenomenon. Ultrasonication treatment creates air bubbles inside the cellulose suspension, gradually releasing energy until they reach their maximum size. At this point, they explode, emitting high pressures and temperatures (of the order of 5000 kPa and 5000 K, respectively) for a relatively short time. The release of energy is sufficient to deconstruct the amorphous domains and cause surface cracks in the crystalline region, thus facilitating the access of enzymes into the cellulose fibers and subsequently enzymatic hydrolysis [56]. 

The SEM technique was used to characterize the validation CNF samples that did not undergo ultrasonic treatment (validation samples A, C, E, and G). The processing of CNF micrographs obtained at 30,000× magnification (not shown) exhibited rough topography with irregular diameter fibrils along the structure. With the software Image J, the CNF diameters were calculated by a series of measurements. The validation samples treated with the optimal enzyme mixture had the smallest CNF diameters, whose average lies between 250 nm to 300 nm; on the other hand, the validation samples hydrolyzed with pure EGU exhibited average CNF diameters between 450 nm to 485 nm. In contrast, Carezyme-treated validation samples showed average CNF diameters lying within 450 nm to 500 nm. Finally, the validation samples hydrolyzed with pure ExG showed average CNF diameters of 550 nm to 585 nm. 

As a result of the CCRD, the average diameter size of the nanocellulose fibers varied between 550 nm and 600 nm; on the other hand, for the validation step, the average diameter size decreased between 250 nm and 300 nm; that is, the average fiber diameter size was reduced by approximately half.

### 2.10. Colby Factor

This study aims to determine an enzymatic synergistic effect using the crystallinity index (CrI) ratio when performing controlled enzymatic modification of nanocellulose with endo-exocellulases. The results obtained to evaluate enzyme synergism were determined using the adapted Colby factor method, in which a ratio of crystallinity indices was used to calculate the Colby factor, as described in Section 3.6. The CrI ratio was used because crystallinity is a feature that influences the rate of cellulose hydrolysis [57] and, consequently, the rate in CNF production. 

Analyzing the CrI (see Table 3) for the hydrolyzed and ultrasonically homogenized samples, there is an increase of approximately 7.8% for the CrI in KCP when it is hydrolyzed under ExG action, 9.2% for hydrolysis under EGU action, and an increase of 11. 7% for hydrolysis under the optimal mix of EGU + ExG, in which the CrI is attributed to chain consumption in the non-crystalline region. For the enzyme synergism analysis, only the validation samples that produced CNF derived from an enzyme mixture and ultrasonic treatment were evaluated, i.e., sample B and sample D. The remaining validation samples, E, F, G, and H, were produced using pure enzymes and not an enzymatic cocktail; therefore, the Colby factor could not be determined for these last four validation samples.

The validation samples hydrolyzed with the optimal enzyme mixture and the Carezyme cocktail showed enzyme synergism by exhibiting Colby factor values > 1, with a maximum synergism of 1.15 for the optimal pure enzyme mixture and 1.06 for the Carezyme cocktail. Therefore, the effect generated by the interactions between enzymes in a mixture is greater than the response that would have been obtained by adding the enzyme effects individually. Thus, the synergistic nature of enzyme interactions for CNF production was demonstrated [29]. 

## 3. Materials and Methods

### 3.1. Materials

The raw material for the CNF preparation was the bleached kraft cellulose pulp (KCP) (Fibria Celulose, Espírito Santo, Brazil). The applied cellulases: Carezyme (enzyme activity ≥1000 U/g, from *Aspergillus* sp.), endoglucanase (≥2 U/g protein, from *Acidothermus cellulolyticus*), exoglucanase (cellobiohydrolase I, 0.13 U/mg, from *Hypocrea jecorina*) and β-glucosidase (≥4 U/g solid, from almonds), were supplied by Sigma Aldrich (São Paulo, Brazil). The sodium citrate buffer (50 mM) used during the enzymatic treatment was prepared from 0,1 M Na_3_C_6_H_5_O_7_ and 0,1 M C_6_H_8_O_7_ so that the pH was between 4.8 and 5.0. All of the chemicals we used were of analytical grade and were supplied by Sigma Aldrich.

### 3.2. Determination of Enzyme Activity

Total cellulase activity (FPase) is determined by the degradation of the paper filter and includes exoglucanase and endoglucanase. This dosage was based on Ghose’s (1987) methodology with modifications. An aliquot of 20 µL of enzyme extract, a 0.6 cm diameter circle of Whatman grade 1 filter paper, and 40 µL of sodium citrate buffer (50 mM, pH 4.8) was added to a PCR plate incubated at 50 °C for 60 min. All of the tests were performed in triplicate. Paper circles were removed to control the enzyme, but other reagents remained. To control the substrate, 60 µL of sodium citrate buffer and a ring of filter paper were added, but only 60 µL of sodium citrate buffer was added to the blank. Following the reaction time, 120 µL of 3,5-dinitrosalicylic acid (DNS) was added to the samples, then placed in a boiling bath for 15 min. After cooling the samples, 20 µL of the reaction mixture was transferred to a 96-well plate containing 180 µL of ultrapure water. A UV spectrophotometer set to 540 nm was used to read the samples. A standard curve was created with a 1 mg/mL glucose solution. 

### 3.3. Determination of Optimal Enzyme Mixture

The endoglucanase (EGU), exoglucanase (ExG), and β-glucosidase (BG) enzyme mixture was optimized using a simplex centroid mixture design varying the total enzyme loading of each experiment between 0.5 and 5.0 U/g, a total solid loading of 25 g/L and considering glucose release (g/L) as the response variable. The quantification of glucose released from each experiment was determined according to the glucose oxidase–peroxidase (GOD-PAP) methodology [58], using a spectrophotometer UV (Shimadzu, UV-1800, Kyoto, Japan).

### 3.4. Preparation of Cellulose Nanofibrils

A central composite rotational design (CCRD) was implemented to produce CNF by varying the total enzymatic loading between 0.5 and 5.0 U/g and the total solids loading between 15 and 50 g/L. 

First, KCP was ground in a Willye-type knife mill (Tecnal, TE-680, Piracicaba, Brazil). The grounded KCP was added into Falcon tubes; then, 15 mL of 50 mM sodium citrate buffer, pH 4.8, was added to each tube, and a different volume of the optimal enzyme mixture was added based on the enzyme loading for each experiment. Enzymatic hydrolysis occurred at 50 °C, pH 4.8, 200 rpm, and 20 h reaction in a shaker (New Brunswick™, Innova^®^44) [59]. After the 20 h reaction, the Falcon tubes were placed in a centrifuge (Thermo Scientific, Waltham, MA, USA, Sorval Lynx 4000) at 3000 rpm. Then, the prepared CNF samples were subjected to successive washings (three times) with deionized water and centrifuged after each washing. The CNF samples were submitted to an ultrasonic bath (Branson, 2510) for 10 min with 42 kHz and 230 W of power. At the end of the ultrasonic bath, the CNF samples were centrifuged for the last time and stored at 4 °C. Table 4 indicates the performance of experiments in the CCRD.

### 3.5. Samples Characterizations

#### 3.5.1. X-ray Diffraction Analysis (XRD)

The XRD patterns for the CNF samples were obtained by an X-ray diffractometer (Rigaku, Japan, Miniflex II) with Cu Kα radiation of 0.15418 nm at 30 kV and 15 mA. The samples were scanned for 2θ from 5° to 40° with the step of 0.02°. The crystallinity index (%) of the CNF samples was estimated by the Segal method:(1)$$\mathrm{CrI}=\frac{I_{200}-I_{\mathrm{AM}}}{I_{200}}\times 100\;\%\;$$
where CrI: crystallinity index (%), I_200_: maximum diffraction intensity corresponding to the crystalline material (cps), and I_AM_: minimum diffraction intensity corresponding to the amorphous material (cps).

#### 3.5.2. Morphology Analysis

The morphologies of the CNF samples were characterized by scanning electron microscopy (SEM) (JSM-7100F, JEOL USA, Inc., Peabody, MA, USA) at the voltage of 2 kV, and those samples did not need any treatment.

#### 3.5.3. Thermogravimetric Analysis (TGA)

The TGA analysis of the KCP and CNF samples was performed with a constant flux of 40 mL/min of nitrogen to ensure that the weight variation was due to thermal degradation. The temperature of the CNF sample was ramped at a constant rate of 20 °C/min at 20–600 °C, an alumina crucible was employed, and the weight loss was measured against the increased temperature on a thermal gravimetric analyzer (Shimadzu, Kyoto, Japan DTG-60H).

#### 3.5.4. Fourier Transform Infrared Analysis (FTIR)

The Fourier transform infrared spectra of the KCP and CNF samples were measured in an FTIR spectrophotometer (Perkin Elmer Frontier, Waltham, MA, USA) in 4000–600 cm^−1^ at a resolution of 4 cm^−1^ in the absorbance mode for 64 scans at room temperature.

#### 3.5.5. Dynamic Light Scattering (DLS)

The KCP and CNF suspensions (0.02% *w*/*v*) were prepared with Milli-Q water and treated with an ultrasonic homogenizer (102C, Branson Digital Sonifier, Danbury, CT, USA) at 400 W and 10% amplitude for 15 min. The particle size distributions of all of the CNF samples were analyzed using a laser diffraction particle size analyzer (Malvern Instruments Ltd., Worcestershire, UK) with ZetaSizer (NanoSeries) software, using Milli-Q water as the solvent (ƞ = 1.333).

#### 3.5.6. Zeta Potential

The surface charges of the KCP and CNF suspensions (0.02% *w*/*v*) were analyzed using Malvern Zetasizer NanoSeries (Worcestershire, UK) at an equilibrium time of 120 s.

### 3.6. Experimental Validation and Colby Factor 

For the evaluation of enzymatic synergism, four new CNF samples were produced in duplicate at 50 °C, pH 4.8, 200 rpm, and 20 h hydrolysis time, using: (i) an optimal mixture obtained with CCRD, (ii) a commercial enzyme cocktail, Carezyme (composed of EGU, ExG, and BG), (iii) pure EGU, and (iv) pure ExG. One sample from each pair of duplicates was subjected to ultrasonic homogenization for 15 min.

The model used to quantify the degree of the synergy of the enzyme mixture, the Colby model [29], was adopted as follows:(2)$$\mathrm{CF}=\frac{\mathrm{Measured}\;\mathrm{CrI}}{\mathrm{Theorical}\;\mathrm{CrI}}$$
where CF: Colby factor, measured CrI: CrI experimentally observed (%), and theoretical CrI: CrI expected from the enzyme mixture (%). It was considered as a theoretical crystallinity index, an average of the CrI values of KCP samples reported in the literature [16,46,47,60].

## 4. Conclusions

This study aimed to optimize synergism enzymatic pretreatment as an eco-friendly method for extracting cellulose nanofibrils (CNF) from cellulose pulp and evaluate the physicochemical and morphological properties of CNF. An optimal enzyme mixture (69% EGU and 31% ExG) was established to produce CNF. XRD indicated that the best CNF sample showed a crystallinity index of 80.9% with the Segal method. SEM determined a CNF diameter trend within 550–600 nm. Subsequently, the optimal enzyme mixture was used ultrasonically to improve the CNF preparation. The SEM results showed CNF with rough topography and fiber diameters within the range of 250–300 nm, so ultrasonic treatment decreases the diameter of the CNF. Moreover, the modified Colby factor formula was implemented to evaluate the degree of enzyme synergism, revealing a maximum value of 15% synergism between the EGU and ExG enzymes. The results derived from the different characterization techniques allow the conclusion that the CNF produced could be used for the manufacture of hydrogels or aerogels because the CNF samples showed moderate to high stability behavior in the colloidal suspension, presenting zeta potential values within the range of ±48 mV and ±52 mV, as well as in food packaging applications and surface coating.

## Figures and Tables

**Figure 1 molecules-28-00948-f001:**
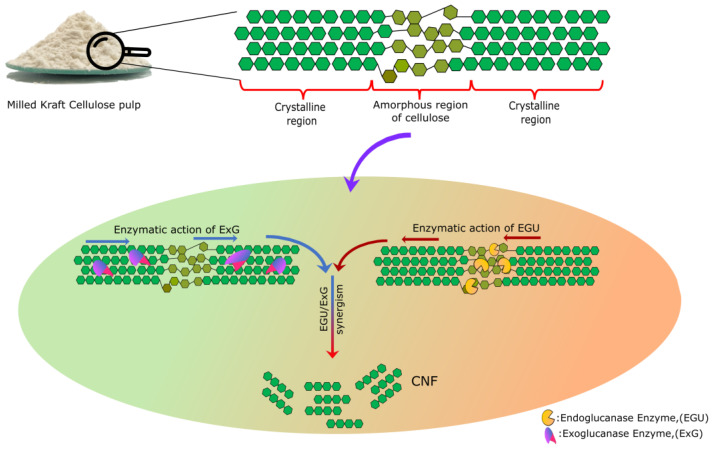
Schematic diagram of the production of CNF derived from the enzymatic hydrolysis of KCP by endo-exocellulase.

**Figure 2 molecules-28-00948-f002:**
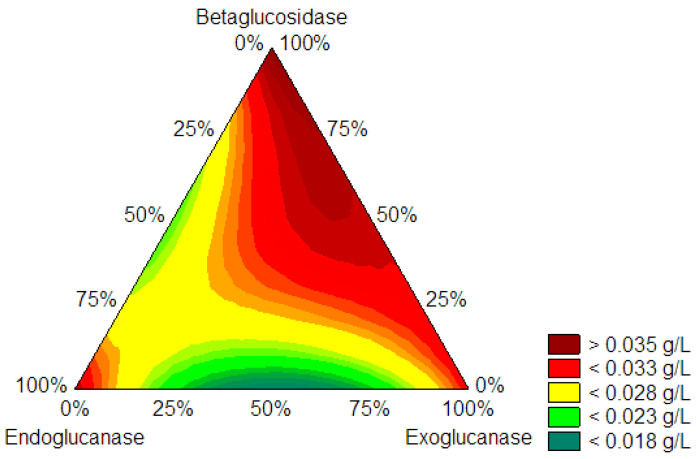
Contour plot of glucose concentration (g/L) as a function of enzyme loading (U/g). The values shown in the legend correspond to the glucose concentration (g/L).

**Figure 3 molecules-28-00948-f003:**
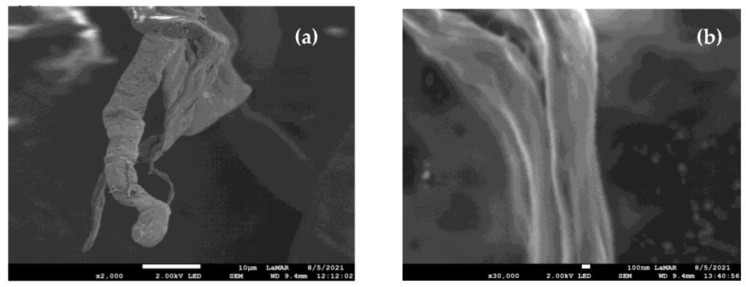
SEM micrographs obtained at the magnification levels of: (**a**) KCP at 2000× And (**b**) CNF sample No. 7 at 30,000×.

**Figure 4 molecules-28-00948-f004:**
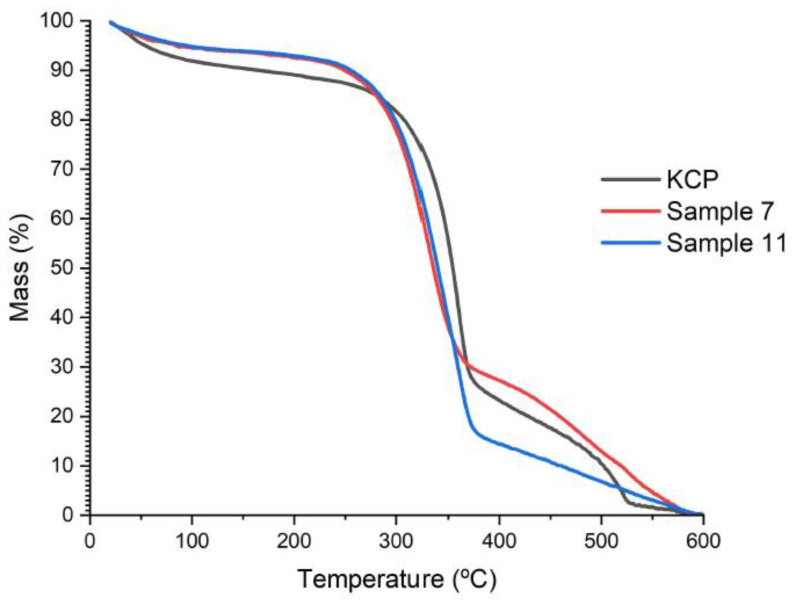
Evaluation of the thermal decomposition of KCP and CNF samples by TGA.

**Figure 5 molecules-28-00948-f005:**
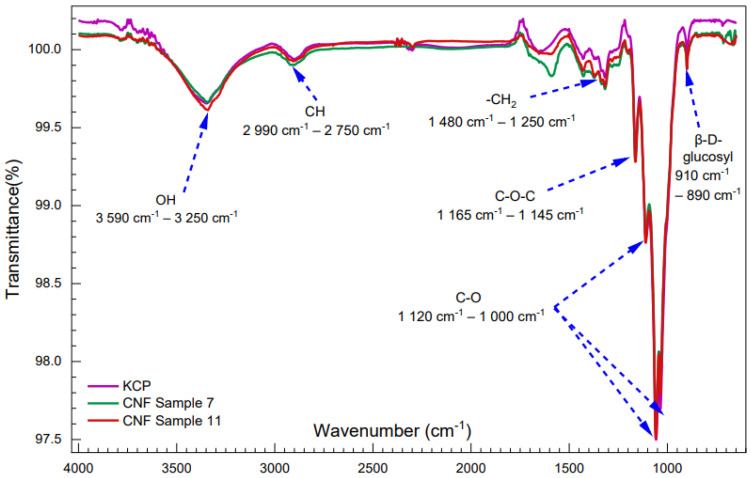
FTIR spectral of the KCP sample (purple curve), CNF sample No. 7 (green curve), and CNF sample No. 11 (red curve).

**Figure 6 molecules-28-00948-f006:**
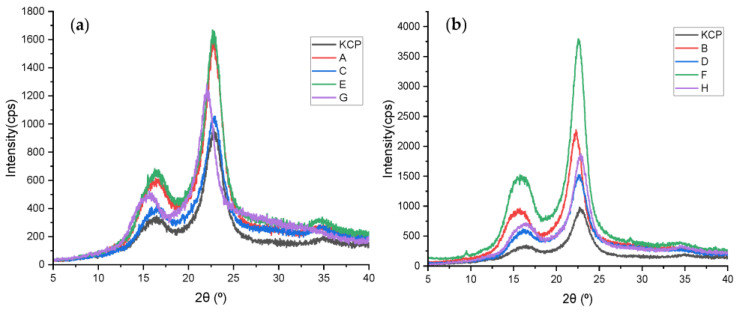
Diffractograms of the validation samples: (**a**) without ultrasonic dispersion treatment and (**b**) with ultrasonic dispersion treatment.

**Table 1 molecules-28-00948-t001:** Crystallinity index of KCP and CNF samples from CCRD.

Sample	KCP	CNF-1	CNF-2	CNF-3	CNF-4	CNF-5	CNF-6
$I_{200}$ (cps)	944.2	1630.8	1143.3	1100	2341.6	1338.3	1608.3
$I_{\mathrm{AM}}\;\left(\mathrm{cps}\right)$	280.8	395	257.5	267	537.5	335	355.8
CrI (%)	70.26	75.78	77.48	75.73	77.05	74.97	77.88
**Sample**	**CNF-7**	**CNF-8**	**CNF-9**	**CNF-10**	**CNF-11**	**CNF-12**	**CNF-13**
$I_{200}\;\left(\mathrm{cps}\right)$	2206.6	1278.3	1367.5	1513.3	2373.3	2041.6	2001.6
$I_{\mathrm{AM}}\;\left(\mathrm{cps}\right)$	420	340	355.8	388.3	528.3	507.5	506.6
CrI (%)	80.97	73.40	73.98	74.34	77.74	75.14	74.69

CrI: crystallinity index (%), I_200_: maximum diffraction intensity corresponding to the crystalline material (cps), and I_AM_: minimum diffraction intensity corresponding to the amorphous material (cps).

**Table 2 molecules-28-00948-t002:** Averages hydrodynamic diameter and zeta potential of KCP and two CNF samples.

Sample CNF	HD (nm)	Zeta Potential (mV)
KCP	509.63	−39.37
7	430.20	−48.47
11	469.70	−52.60

**Table 3 molecules-28-00948-t003:** Crystallinity indices of validation samples.

Sample Code	Subjected to Ultrasound Treatment	CrI (%)	Enzyme/s Used
KCP	No	70.26	None
A	No	74.78	EGU + ExG (Optimal mixture)
B	Yes	78.45	
C	No	71.93	Carezyme cocktail
D	Yes	72.27	
E	No	74.63	Pure EGU
F	Yes	76.68	
G	No	72.13	Pure ExG
H	Yes	75.80	

**Table 4 molecules-28-00948-t004:** Variables and levels used in the central composite rotational design (CCRD).

Sample	CNF-1	CNF-2	CNF-3	CNF-4	CNF-5	CNF-6	CNF-7
Enzyme loading (U/g)	1.16	4.34	1.16	4.34	0.50	5.00	2.75
Solids loading (g/L)	20	20	45	45	33	33	15
**Sample**	**CNF-8**	**CN-F9**	**CNF-10**	**CNF-11**	**CNF-12**	**CNF-13**	
Enzyme loading (U/g)	2.75	2.75	2.75	2.75	2.75	2.75	
Solids loading (g/L)	50	33	33	33	33	33	

## Data Availability

Not Applicable.

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
