# Peer review of "Endo-Exoglucanase Synergism for Cellulose Nanofibril Production Assessment and Characterization"

_molecules, 2023, doi:10.3390/molecules28030948_

Round 1

Reviewer 1 Report

COMMENTS ON THE ARTICLE

·    -  There are formal mistakes in the article: for example, sometimes the unit of measurement is written next to the number (line 135; line 253) and sometimes separately (line 141). “No.” (line 225), etc. It should be checked and corrected all these issues throughout the article.

·   -  In Figure 1, the values shown in the legend, what do they correspond to? What are their units?. These aspects should be clarified.

·       -  Lines 11-12: These sentences need to be revised as they do not make sense with this structure.

·     -   Lines 16-19: from "The average" to "dispersion", at least, one verb is missing for the paragraph to make sense.

·  - Lines 64-68: the word "between" has been overused in this paragraph. Paragraph might be redefined.

·    - Lines 75-77: from "therefore" to "recommended" the paragraph should be reworded as it makes no sense.

·   - Lines 93-101: the effect of some enzymatic hydrolysis parameters on cellulose crystallinity is discussed in this article, but the advantages of obtaining, after enzymatic hydrolysis, nanocellulose with a high level of crystallinity have not been commented. These advantages should be included in the introduction section in order to adequately justify this study.

·         Lines 109-116.

A) If the objective is to obtain nanocellulose (fibers), what is the point of studying the action of exoglucanases, whose main objective is to release cellobiose, which would not support the nanocellulose production? Justify this question adequately.

According to the action of the types of enzymes studied, which are well known, it could be intuited that the presence of exo-glucanases and β-glucosidases would hinder the generation of nanocellulose, which questions the effectiveness of the experimental design requested.  Has this aspect been considered beforehandIt. Is possible that the study of synergism between endoglucanases and exoglucanases in enzymatic hydrolysis makes more sense if the aim is to obtain a high concentration of monosaccharides rather than to obtain nanocellulose.

 B)    It is concluded that the optimum values correspond to 69 % EGU and 31 % ExG, because they lead to a lower glucose concentration, but what about the generation of cellobiose, due to the action of ExG? Cellobiose has not been quantified and could lead to low yields of nanocellulose. Not considering the concentration of cellobiose formed could have led to inaccurate results for this optimization. Why has the concentration of cellobiose not also been minimized as a possible response? Has this fact been considered?; Can any comment be included in the article?

C)      Should lower concentrations of exoglucanase have been tested to increase the process yield? so some comments should be made in the article in this respect.

D)      Why was a solid loading of 25 g/L chosen?

 E)      Why does the loading of each enzyme vary between the limits 0.5 and 5.0 U/g?. Are these ones the limits for each type of enzyme (exoglucanase, endoglucanase and β-glucosidase)? How many assays have been performed in total? All these aspects are not clear from the article so they should be further clarified in the article.

· Lines 129-130. It should be clearly specified in the article how this optimization has been obtained to conclude that 0.974 U/g enzyme load and 15 g/L solids load are the most appropriate parameters values. Are there any mathematical models of adjustment? So The models should be further clarified in the article.

· Line 130. What is the meaning of the symbol "?" in this article? Is it the p-value? Define the parameter in the article.

·   Lines 142-146. This information should be moved to the "Introduction" section as it does not make sense to include it in this section.

· Lines 148-149. The information given is not correct. Comments on Figures S1-S13 should have been made

· Include, in Table 1, just the information on the intensities I200 and IAM from which these values have been calculated. It is the basis for decisions in this article and the Figures provided as additional material do not show data sufficiently.

· Lines 158-164. Why is Segal's method used to determine the crystallinity index if it is considered, in this article, as not a very precise method? Some positive justifications for the use of this method should be pointed out in this paragraph.

· Lines 169 and 170. Put et al. in italics. Correct throughout the article (specially in Bibliography).

·   Lines 171. There is an error: apart from No 7, sample No 11 has not the highest CrI but No 6 has. Why was sample No 6 not characterized instead of No 11? This fact affects the rest of the research that has been carried out.

·    Lines 180-182. It is not clear what is meant with these sentences. Clarify better in the article.

·  Lines 184-190: the information shown in the text correspond to CNC and not to CNF. It seems that, according to the data offered in lines 180-182, the results are close to cellulose in the form of nanocrystals. Would these data therefore be incompatible with obtaining nanocellulose microfibrils?

· Line 195. Dynamic light scattering (DLS) technique has not been explained in methods. It should be reported.

· Lines 223-224. According to the zeta potential values shown, the nanocellulose obtained would constitute a stable colloidal suspension in aqueous solution. Please, explain clearly in the article the advantages that this fact would entail for nanoencapsulation, at least for drug delivery, as this is the focus of this research.

·  Line 287: It is not really understood in this section how the validation has been carried out. Explain in detail and accurately what the validation consists of.

·  Lines 288-289. What program was used to carry out the optimization? How has the optimization been carried out and what mathematical models have been obtained? It is necessary to give much more information about the optimization process that has been carried out, and whether the variables considered are significant. Is there a lack of fit in the models?. Comment on the article in this respect

·     Lines 314-316: Table 3 shows little difference for CrI between the use of EGU + ExG and EGU. The option of using a mixture of EGU+ExG and not only EGU may not be as evident. This could be due to experimental errors in the process. Would the tests have been carried out in duplicate? Should the standard deviation have been calculated?

·  Lines 356-359. This paragraph is not understood, nor is an explanation given for the comment. Rectify and comment again.

· Lines 366-368. This is true, but, in this study, the crystallinity index of CNFs to be subjected to enzymatic hydrolysis has not been calculated and studied, as this is not the objective. What is the point of this comment in the article? so perhaps this should be reflected upon and corrections should be made.

· Lines 369-377: this is information that should be considered in the "Introduction" section.

·  Line 394: Materials and methods should be placed before the discussion of results.

· Line 421: Which program was used to determine the optimal enzyme mixture? Explain in this section exactly how this optimization was carried out.

·   Lines 441-442. Why for 10 minutes, 42 kHz and 230 W?

·   Line 447. Table 4 is meaningless. The information contained in this table can be commented on in lines 430-432 to complete the information in the paragraph. Delete this table.

·     Line: 578: typographical error. It is advisable to check the bibliography.

·     Line 612: typographical error. It is advisable to check the bibliography.

Author Response

Please see the attachment for the point-by-point response to the reviewer’s comments and corrected manuscript. 

Reviewer 2 Report

The paper “Endo-exoglucanase synergism for cellulose nanofibril production assessment and characterization” aims to optimize synergism enzymatic pretreatment as an eco-friendly method for extracting cellulose nanofibrils (CNF) from cellulose pulp and evaluate the physicochemical and morphological properties of CNF. An optimal enzyme mixture was established to produce CNF. Various methods were used to characterize the CNF. Finally, synergistic interactions between the enzymes involved in nanocellulose production were demonstrated. This paper suggests a better alternative for CNF material production. However, some major issues should be addressed before this manuscript is suitable for publication in Molecules.

(1) The grammar and language should be improved in the manuscript. There are places where the narrative is rather difficult to follow, which slightly obscures the description or statement that is being made. In the revised version, I would recommend additional proof-reading of the manuscript.

(2) The introduction part should be improved to be more logical. As we know, many researchers have reported the preparation of CNF by enzymolysis approach. Thus, it is better for you to make a comparation include cost, process, dimension, and so on.

(3) What’s the yield of CNF by this method ?

(4) Detailed mechanism of how about the CNF was separated from cellulose pulp by enzymatic pretreatment should be provided, and a schematic diagram is suggested.

(5) Many important literature is suggested to cite, for example, “Comparative study on properties of nanocellulose derived from sustainable biomass resources”, “Cellulose nanofibrils manufactured by various methods with application as paper strength additives”, “Isolation and rheological characterization of cellulose nanofibrils (CNFs) produced by microfluidic homogenization, ball-milling, grinding and refining”, “Cellulose nanofibrils (CNFs) produced by different mechanical methods to improve mechanical properties of recycled paper”, “Properties of cellulose nanofibril produced from wet ball milling after enzymatic treatment vs. mechanical grinding of bleached softwood kraft fibers”, “Green synthesis of bacterial cellulose via acetic acid pre-hydrolysis liquor of agricultural corn stalk used as carbon source”, and so on.

Author Response

(The authors gave the same response as above.)

Round 2

Reviewer 1 Report

In general, the discussion in section 2.1. "Determination of the optimal model of the mixture of enzymes" should be improved based on some comments made in the first review. Specifically, considering the aforementioned epigraph, it is expected that the model (mathematical equation) of adjustment of the independent variable (glucose concentration) will be given.

Notes: 

Line 2. Remove the double period.

Line 56. The meaning of these acronyms “KCP” has not been previously established.

Line 156. End point is missing.

etc.
